# Examining Romosozumab Adherence and Side Effects in Osteoporotic Patients After Surgical Fracture Fixation: A Comparative, Descriptive, and Hypothesis-Generating Study with Non-Fractured Controls

**DOI:** 10.3390/diseases13050148

**Published:** 2025-05-11

**Authors:** Amarildo Smakaj, Umberto Tarantino, Riccardo Iundusi, Angela Chiavoghilefu, Lorenzo Abbondante, Chiara Salvati, Chiara Greggi, Elena Gasbarra

**Affiliations:** 1Department of Clinical Sciences and Translational Medicine, “Tor Vergata” University of Rome, 00133 Rome, Italy; amarildo.smakaj@gmail.com (A.S.); umberto.tarantino@uniroma2.it (U.T.); angela.chiavo@hotmail.com (A.C.); lorenzo-abbondante@hotmail.it (L.A.); chiarasalvati95@yahoo.it (C.S.); chiara.greggi@gmail.com (C.G.); elena.gasbarra@ptvonline.it (E.G.); 2Department of Orthopaedics and Traumatology, “Policlinico Tor Vergata” Foundation, 00133 Rome, Italy; 3Department of Biomedicine and Prevention, “Tor Vergata” University of Rome, Via Montpellier 1, 00133 Rome, Italy; 4Dipartimento di Scienze Chirurgiche, Università Cattolica Nostra Signora del Buon Consiglio, 1000 Tirana, Albania

**Keywords:** osteoporosis, fragility fractures, Fracture Liaison Service (FLS), Romosozumab, bone metabolism, post-operative osteoporosis therapy

## Abstract

Objectives: The study aims to evaluate adherence to Romosozumab treatment in osteoporotic patients after surgical fracture fixation and compare side effects with non-fractured controls on the same therapy. Methods: This retrospective case–control study was conducted at the Orthopaedic Department of Policlinico Universitario di Roma “Tor Vergata”, following the principles of the Declaration of Helsinki. It included postmenopausal women aged over 60, with the case group receiving Romosozumab after fracture fixation, and the control group consisting of women on Romosozumab therapy without fracture fixation. Exclusion criteria included psychiatric conditions, contraindications to Romosozumab, high-energy trauma, or other bone metabolism disorders. Data on fractures, surgeries, FRAX (Fracture Risk Assessment Tool) scores, BMD (Bone Mineral Densit) values, and follow-up details were collected. Side effects, including nasopharyngitis and severe events like hypocalcemia, stroke, and myocardial infarction, were recorded. Adherence was assessed via pharmacy records and patient interviews during routine clinical follow-up visits. Statistical analysis was performed using descriptive statistics, t-tests, and chi-square tests. Results: The study included 25 patients, with 12 in the surgical group and 13 in the conservative treatment group. The surgical group had a mean age of 67.3 years and a follow-up of 374 days, while the conservative group had a mean age of 76.4 years and a follow-up of 287 days. The surgical group underwent various fracture treatments, including femoral, humeral, and distal radius fractures, while the conservative group was treated with immobilization. There were no significant differences in FRAX scores or BMD values between the two groups. Vitamin D levels increased significantly in both groups after supplementation, but parathyroid hormone levels showed no difference. No new fractures occurred, and surgical patients had no delayed union or nonunion, though two had superficial wound infections. Conclusions: Both groups adhered well to Romosozumab therapy, with no severe side effects; minor side effects included myalgia in the surgical group and shoulder arthralgia in the conservative group. Romosozumab is well-tolerated and adherent in osteoporotic patients after osteosynthesis surgery, with adverse events similar to non-fractured individuals. While the study design is appropriate, multicenter trials would improve the sample size and allow for subgroup analysis based on fracture type and demographics.

## 1. Introduction

Osteoporosis affects 5.5% of individuals in the overall general population of the European Union: 6.6% of men and 22.1% of women aged 50 years or more [1]. Incidence in the EU is 21/100,000 in the population at risk (people aged 50 years or more) and total fragility fractures in the five largest European countries plus Sweden are estimated to increase from EUR 2.7 million in 2017 to EUR 3.3 million in 2030 [2]. Costs for healthcare systems are massive, with EUR 36.5 million spent in the EU in 2019 alone [3].

The available data indicate that individuals who have experienced fragility fractures are at an elevated risk of sustaining additional fragility fractures over time [4,5]. Therefore, it is crucial to address osteoporosis in order to prevent the occurrence of further fractures [6]. The multidisciplinary approach through the implementation of the Fracture Liaison Service (FLS) has made it possible to reduce the risk of secondary fragility fracture compared to the non-FLS population at 1-year follow-up [7,8], and this outcome was maintained for 2 or more years with a moderate degree of evidence [7,9].

Moreover, the development of new drugs has played a key role in improving the therapeutic approach to osteoporosis [10]. One of the most recently introduced drugs include Romosozumab, a monoclonal antibody that falls in the anabolic category of bone metabolism [11]. This monoclonal antibody binds and inhibits sclerostin which is a negative regulator of bone formation by Wnt/β—catenin signaling downregulation [11,12]. However, it also has a double effect on bones, causing increased bone formation and decreased bone resorption [13]. Until now, Romosozumab has been suggested for the treatment of osteoporosis in post-menopausal women at high risk of fractures [14,15]. It has been shown to exert greater increases in BMD and fracture risk reduction in treatment-naïve patients compared to those previously exposed to antiresorptive therapies [16]. According to the product label, known adverse events include arthralgia, myalgia, headache, and nasopharyngitis, while rare but serious concerns such as cardiovascular events (myocardial infarction and stroke) have also been reported [15]. Recent meta-analyses have attempted to identify predictors of therapeutic response and adverse events, highlighting the potential influence of baseline bone turnover markers, prior antiresorptive therapy, and cardiovascular risk factors [17]. However, there remains a gap in the literature regarding the use of Romosozumab in the immediate post-operative setting following fracture fixation.

To our knowledge, there are no studies in the literature on the use of Romosozumab in the post-operative period following fragility fracture reduction and fixation surgery. In fact, there is a lack of clear guidance on the use of this drug in the post-operative setting, including its safety profile, patient adherence to the therapy, and the optimal timing for prescription.

Therefore, the aim of the study is to assess the level of adherence to Romosozumab treatment when prescribed at the time of discharge in osteoporotic patients who underwent surgical fracture fixation. Moreover, we want to compare the incidence and nature of side effects between this group of patients and non-fractured controls under Romosozumab therapy.

## 2. Material and Methods

This is a retrospective analytical study with a case–control design, conducted in accordance with the principles of the Declaration of Helsinki. As it is an observational study, the local Ethics Committee has confirmed that no ethical approval is required because it was based solely on anonymized data. Patients were recruited from the Orthopaedic Department of Policlinico Universitario di Roma “Tor Vergata”. The case group included postmenopausal women aged 60 years or older who received Romosozumab therapy at the time of hospital discharge, following fracture fixation or joint replacement surgery for low-energy fragility fractures. The control group consisted of postmenopausal women in the same age range who were undergoing Romosozumab treatment but had no history of surgical fracture fixation. All control group patients were recruited through the hospital’s Fracture Liaison Service (FLS) and selected based on the same inclusion and exclusion criteria as the case group, except for the surgical intervention. All patients received Romosozumab at the standard dose of 210 mg administered subcutaneously once a month for 12 months. Eligibility for Romosozumab was defined according to the Italian Medicines Agency (AIFA) Nota 79, which limits prescription to patients at a very high risk of fracture. Specifically, treatment was indicated in women with a history of multiple fragility fractures, or with a recent severe fragility fracture (such as vertebral or femoral) occurring within the last 24 months or with a T-score ≤ −3.5 at the lumbar spine or femoral neck. Prescription was also allowed in cases of documented failure or intolerance to previous antiresorptive therapies. Patients were excluded from the study if they sustained fractures due to high-energy trauma or had bone metabolic disorders other than osteoporosis, including Paget’s disease, osteomalacia, or hyperparathyroidism. Additional exclusion criteria were the presence of psychiatric conditions—such as major depression—that could interfere with treatment adherence or self-reporting of outcomes as well as contraindications to Romosozumab therapy. These included known hypersensitivity to the drug, uncorrected hypocalcemia prior to treatment or a history of major cardiovascular events such as stroke or myocardial infarction within the previous 12 months. Patients with incomplete or insufficient clinical and radiographic documentation were also excluded from the analysis. Additionally, a personal history of a bone metabolism disease other than osteoporosis was regarded as an exclusion criterion. Patients were included in the study only if they had a follow-up period of at least twelve months after starting anti-osteoporotic therapy.

Hospital and outpatient records were systematically reviewed to collect demographic information, fracture type specifics, details on surgical procedures, start time of the therapy, pre-operative FRAX scores, pre-operative BMD values, twelve-month follow-up BMD values as well as pre- and twelve-month follow-up values for PTH and Vitamin D. Delayed union and nonunion were reported. Individuals with incomplete medical records that hinder comprehensive data collection were excluded from the study. All patients were enrolled in the active Fracture Liaison Service (FLS) at the Policlinico Tor Vergata.

Side effects were also recorded. Common side effects included nasopharyngitis, arthralgia, headaches, and myalgia. Hypocalcemia, stroke, and myocardial infarction were considered severe side effects. Pharmacy records and patients’ interviews were used to assess medication adherence in terms of compliance and persistence.

The data underwent analysis for descriptive statistics, with mean or median values applied for continuous variables and frequency distribution percentages for categorical variables. Student’s t-test was employed to assess distinctions between case and control groups concerning continuous data, while the chi-square test was utilized for the examination of categorical variables; *p*-value < 0.05 was considered statistically significant. Since this was a retrospective study, no a priori sample size calculation was performed. The total number of patients included reflects all eligible individuals treated during the study period who met inclusion criteria. Given the limited sample size, post hoc power analysis was considered methodologically inappropriate, as it would be based on unsupported assumptions about the effect size and event rates. This study was therefore designed as an exploratory, hypothesis-generating investigation aiming to describe real-world patterns of adherence and safety.

## 3. Results

The study population consisted of 25 individuals enrolled in 2023. This timeframe was chosen because Romosozumab only became routinely available for clinical use in our setting starting in early 2023, allowing for standardized treatment protocols and complete electronic documentation. Twelve participants received surgical treatment, while the remaining 13 underwent conservative therapy. The mean age of the surgical treatment group was 67.3 ± 14.9 years, which differed from the 76.4 ± 7.9 years observed in the conservative treatment group. The average follow-up duration for the surgical group was 374.22 ± 102.31 days, in contrast to 386.73 ± 90.60 days for the conservative treatment group (Table 1). Major comorbidities are reported in Table 1. The surgical cohort underwent a total of nine procedures for femoral fractures, two for proximal humeral fractures, one for a distal radius fracture, and one for tibia and fibula fractures (Table 2). Three of the femoral fractures were managed with total hip arthroplasty, while the remaining two received intramedullary nail fixation. The two proximal humeral fractures were treated using different approaches: one with a reverse total shoulder replacement and the other with an intramedullary nail. The distal radius fracture was addressed with plate and screw fixation, and the tibia and fibula fractures were treated using a combination of techniques. In the control group, the fractures were conservatively managed with appropriate immobilization using braces, including six vertebral fractures, three proximal humeral fractures, three distal radius fractures, and two metatarsal fractures (Table 2). The distribution of prior anti-osteoporotic therapy among patients varied between the surgical and conservative groups. The majority of patients in both groups had no previous therapy, with seven in the surgical group and twelve in the conservative group. Vitamin D and calcium supplementation was the most common therapy among those treated, reported by three patients in the surgical group and five in the conservative group. Bisphosphonates were used by one patient in the surgical group and two in the conservative group, while denosumab was reported by one patient in the surgical group and none in the conservative group. No patients had received teriparatide or other therapies (Table 2).

The therapy start time was 12.4 ± 3.0 days after the fracture for the surgery group, compared to 6.42 ± 1.6 days for the control group, with a statistically significant difference between the two groups. The mean FRAX score was 29.5 ± 3.1 for the surgically treated group, whereas it was 28.7 ± 2.6 for the conservatively treated group, with no statistically significant differences between the two groups (Table 1). All patients received appropriate supplementation of vitamin D and calcium in cases where their daily intake through diet was deemed insufficient. For the surgery group, the pre-operative femoral neck BMD value was −2.9 ± 0.3, which improved to −2.6 ± 0.3 at the twelve-month follow-up. At the L1–L4 level, the pre-operative value was −2.9 ± 0.3 and slightly improved to −2.8 ± 0.4. In the control group, the femoral neck BMD values changed from −2.8 ± 0.3 at baseline to −2.5 ± 0.4 at the twelve-month follow-up, while the L1–L4 values shifted from −2.8 ± 0.3 to −2.5 ± 0.4. Data show no statistically significant differences between the two groups in either femoral or L1–L4 BMD values at pre-operative and twelve-month follow-up (Table 1).

The mean vitamin D levels in the surgically treated patients increased from 22.7 ± 3.9 ng/mL post-operatively to 37.9 ± 3.8 ng/mL at the twelve-month follow-up, following appropriate supplementation after discharge. Similarly, the mean vitamin D levels in the conservatively treated patients with fragility fractures rose from 23.7 ± 4.0 ng/mL post-operatively to 36.2 ± 3.1 ng/mL at the twelve-month follow-up, also after receiving appropriate supplementation (Table 1). Statistical analysis revealed a significant increase in vitamin D levels after supplementation in both groups. However, no differences were observed in the parathyroid hormone levels between the two groups. During the follow-up period, no new fractures were reported. The patients who underwent osteosynthesis did not experience any delayed union or nonunion. Two post-operative complications involving superficial wound infections were observed in the surgery group

Analysis of the hospital pharmacy records indicated that patients in both groups consistently collected their prescribed medications (Table 1). Additionally, interviews with the patients revealed that they reported adhering to the prescribed subcutaneous administration of Romosozumab. No severe side effects were reported in either group. Among the surgically treated patients, two reported myalgia, while one patient in the conservatively treated group reported bilateral shoulder arthralgia (Table 3).

## 4. Discussion

Our findings indicate that both the surgically treated patients and the conservatively treated control group adhered well to the prescribed Romosozumab therapy. The hospital pharmacy records and patient interviews during routine clinical follow-up visits confirmed consistent medication collection and subcutaneous administration. This suggests that Romosozumab is generally well-received and that patients in both groups followed the prescribed regimen. The high adherence rates observed in this study are consistent with previous reports that highlight the importance of patient education and follow-up care in ensuring compliance with osteoporosis therapies, including biologic treatments like Romosozumab. The high adherence to treatment can also be attributed to the inclusion of patients in the Fracture Liaison Service (FLS) [18,19], which provided structured follow-up and support throughout the treatment process. However, a great variety of agents are currently available for the treatment of osteoporosis. Clinical trials have demonstrated that these agents show varying degrees of efficacy, ranging from 30% to 70%, in reducing risk for fragility fracture [20]. Compliance and persistence are challenging to measure accurately in the real-world setting. In controlled clinical trials, patients are constantly monitored, and rates of compliance and persistence are generally high [21]. In daily practice, however, evaluation of compliance and persistence can be achieved only by patient self-reports, which show varying degrees of reliability, or through analysis of medical and information claims databases, such as those of large managed care organizations or national health systems. Ultimately, once a prescription has been filled, the actual fate of the medication lies with the individual. Despite these limitations, retrospective and observational analyses currently represent the most useful means by which to estimate real-world medication usage. Such studies indicate that compliance and persistence with osteoporosis therapies in daily practice are suboptimal [22]; indeed, it has been estimated that among individuals at risk for fracture, 50% are poorly compliant or poorly persistent within 12 months of initiating treatment [23]. The relation between poor compliance or persistence and fracture rate is difficult to quantify. The benefits afforded by treatment are likely to depend on the level of medication that is actually taken, how regularly the medication is taken, and the length of time that the medication is taken at a minimally effective dose [24]. The consequences of poor adherence—primarily fewer avoided fractures than is optimal—are likely to be far reaching [25]. For individual patients, a reduction in overall health-related quality of life might be expected secondary to an associated increase in morbidity and mortality. For healthcare systems, the increase in morbidity and mortality can be expected to increase the costs associated with managing fractures. Even without taking fracture rates into consideration, poor adherence results in investment loss for the healthcare systems that are paying for medications that, through suboptimal use, may not be providing their predicted benefit. In addition, there may be costs involved in managing side effects of an osteoporosis medication, even when it is not taken in a manner likely to provide benefit [26]. Analyzing the available literature there are many reviews that examine persistence and compliance in relation to fracture rates [27,28]. Interpretation of the findings is, however, limited by differences in the methodologies used by the individual studies: although most studies considered the impact of confounding factors—age, sex, prior treatments, bone mineral density, etc.—on fracture risk, this approach was not uniformly applied, and in a limited number of studies no adjustment was made.

The study also assessed bone mineral density (BMD) using BMD values at the femoral and L1–L4 levels. While both groups showed slight improvements in bone density over the twelve-month follow-up period, no statistically significant differences were observed between the surgically treated and conservatively treated groups. The lack of significant difference in BMD values may reflect the relatively short follow-up period of twelve months, as more substantial changes in bone density are often observed over longer treatment durations. Previous studies on Romosozumab have demonstrated significant improvements in BMD after one year of therapy, particularly in post-menopausal women with osteoporosis [14,16,29,30]. Therefore, it would be useful for future studies to extend the follow-up period to better assess the long-term effects of Romosozumab on bone density in both post-fracture surgical and conservative treatment settings. Moreover, a significant increase in vitamin D levels was observed in both groups following appropriate supplementation, which is critical in the management of osteoporosis and fracture healing. Vitamin D is essential for calcium absorption and bone mineralization, and its supplementation in osteoporotic patients has been shown to improve bone health and reduce the risk of fractures. The results highlight a notable proportion of patients who had not received any prior anti-osteoporotic therapy, despite their risk of fracture. This underscores the need for improved screening and prevention strategies in osteoporosis management. Among those treated, the predominance of vitamin D and calcium supplementation suggests that basic preventative measures are prioritized over pharmacological therapies. The low utilization of bisphosphonates and denosumab may reflect either patient selection criteria, limited access, or under-prescription of these more targeted therapies. The absence of anabolic agents like teriparatide further indicates a potential gap in the use of advanced treatment options for osteoporosis. Emerging evidence highlights the utility of bone turnover markers such as tartrate-resistant acid phosphatase 5b (TRACP-5b), procollagen type 1 N-terminal propeptide (P1NP), and bone-specific alkaline phosphatase (BAP) in predicting the early anabolic response to Romosozumab, particularly in patients with severe osteoporosis or high baseline turnover rates [31]. These biomarkers can offer dynamic insights complementary to DXA-based BMD measurements, and their early modulation may correlate with long-term gains in skeletal mass and strength [32]. Nonetheless, their use was not feasible in our retrospective cohort, representing a limitation of the study, as efficacy assessment was based solely on densitometric and clinical outcomes.

The therapy start time was significantly delayed in the surgical group compared to the conservative group, likely due to the need for preoperative assessments and planning before surgery. Despite the delayed therapy start in the surgical group, both cohorts showed similar improvements in clinical outcomes, highlighting the effectiveness of both treatment approaches.

The safety profile of Romosozumab in both groups was generally favorable. No severe side effects such as hypocalcemia, stroke, or myocardial infarction were reported, and only mild, transient side effects were observed. In the surgically treated group, two patients reported myalgia, while one patient in the control group experienced bilateral shoulder arthralgia. These mild side effects are consistent with those observed in clinical trials of Romosozumab, where musculoskeletal pain and arthralgia are among the most common adverse events, although these tend to be self-limiting and resolve with continued therapy [13,15]. The absence of severe side effects and the minimal occurrence of musculoskeletal complaints suggest that Romosozumab is well-tolerated in both post-fracture surgical and conservative settings.

The surgical cohort did not experience any cases of delayed union or nonunion, which is noteworthy, as these complications can significantly impact recovery in osteoporotic patients. The favorable outcomes in terms of fracture healing may be attributed to the combination of surgical stabilization and Romosozumab therapy, which has been shown to promote bone formation and improve fracture healing in osteoporotic patients [33,34]. However, a randomized controlled trial did not find a significant difference in its primary endpoint, failing to demonstrate acceleration of fracture healing in femoral fractures [35]. Furthermore, the absence of new fractures in both groups during the follow-up period highlights the efficacy of Romosozumab in reducing the risk of secondary fractures, a key concern in osteoporosis management. These positive results suggest that the prescription of Romosozumab at discharge following an osteoporotic fracture could be considered a good clinical practice.

While this study provides valuable insights into the use of Romosozumab in post-operative fracture management, there are several limitations that should be addressed in future research. First, it is a retrospective study, which means it looks back at past data rather than using a prospective, controlled design. This retrospective approach can introduce biases, such as selection bias or incomplete data. Second, the sample size is relatively small, which may limit the ability to generalize the results to a wider population and reduce the statistical power to detect significant differences. Additionally, the patients prescribed the treatment had varied characteristics, such as age, other health conditions, or severity of the condition. These individual differences could affect the treatment outcomes. The extremely small sample size did not allow for the use of statistical models. This constraint limited our ability to perform causal inference. Therefore, the findings of this study should be interpreted as descriptive and hypothesis-generating, providing groundwork for future prospective investigations with adequate statistical power and control for confounding factors. Furthermore, the lack of long-term follow-up data means the long-term effectiveness and safety of the intervention are unknown. Finally, potential issues with how the data were reported or collected could impact the reliability of the findings. To address these limitations, future studies should employ a prospective, randomized controlled design with larger and more similar patient groups.

## 5. Conclusions

Our findings suggest that Romosozumab is generally well tolerated and shows good adherence in osteoporotic patients undergoing osteosynthesis surgery. Its prescription at discharge appears to be feasible and did not result in a higher incidence of adverse events compared to non-fractured patients in our cohort. However, due to the retrospective design and limited sample size, these results should be interpreted with caution. Larger, prospective multicenter studies are needed to confirm these observations, better define safety and adherence profiles, and allow for subgroup analyses based on fracture type and patient characteristics.

## Figures and Tables

**Table 1 diseases-13-00148-t001:** Demographic and clinical data.

		Surgical Group	CTRL Group	*p* Value
Number of Patients		12	13	-
Age (years)		67.3 ± 14.9	76.4 ± 7.9	>0.05
Follow-up (days)		374.22 ± 102.31	386.73 ± 90.60	-
Doses administered		100% (12)	100% (12)	-
TST *		12.4 ± 3.0	6.42 ± 1.6	<0.001
Vit. D (ng/mL)				
	Pre op	22.7 ± 3.9	23.7 ± 4.0	-
	Post op	37.9 ± 3.8	36.2 ± 3.1	-
PTH				
	Pre op	78.1 ± 29.6	59.8 ± 14.4	-
	Post op	65.8 ± 13.4	60.4 ± 14.0	-
T-score femoral neck				
	Pre op	−2.9 ± 0.3	−2.8 ± 0.3	-
	Post op	2.6 ± 0.3	−2.5 ± 0.4	-
T-score L1-L4				
	Pre op	−2.9 ± 0.3	−2.8 ± 0.3	-
	Post op	−2.8 ± 0.4	−2.5 ± 0.4	-
FRAX		29.5 ± 3.1	28.7 ± 2.6	-

* Therapy start time: indicate how many days after the surgical procedure the patients started the therapy with Romosozumab.

**Table 2 diseases-13-00148-t002:** Details on previous antiosteoporotic therapy, type of fracture, and comorbidities.

		Surgical Group	Conservative
Previous antiosteoporotic therapy			
	No therapy	7	6
	Vit. D + calcium	3	5
	Bisphosphonate	1	2
	Denosumab	1	-
	Theripathide	-	-
	Other	-	-
Type of fracture			
	Femoral	9	-
	Proximal humerus	2	3
	Distal radius	1	3
	Tibia and fibula	1	-
	Vertebral	-	6
	Metatarsal	-	2
Comorbidities			
	Hyperthention	7	8
	Diabeties	4	6
	Tumor	2	3
	Dyslipidemia	3	4
	Others	2	3

**Table 3 diseases-13-00148-t003:** Complications.

	Surgical Group	CTRL Group
Nasopharyngitis	-	2
Arthralgia	4	3
Headaches	2	1
Myalgia	3	2
Hypocalcemia	-	-
Stroke	-	-
Myocardial infarction	-	-
Others	-	-

## Data Availability

Data will be made available upon reasonable request.

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
