# Peer review of "Examining Romosozumab Adherence and Side Effects in Osteoporotic Patients After Surgical Fracture Fixation: A Comparative, Descriptive, and Hypothesis-Generating Study with Non-Fractured Controls"

_diseases, 2025, doi:10.3390/diseases13050148_

Round 1
Reviewer 1 Report
Comments and Suggestions for Authors
The paper named “Examining Romosozumab Adherence and Side Effects in Osteoporotic Patients After Surgical Fracture Fixation: A Comparative Study with Non-Fractured Controls” addresses a very interesting topic based on the known anabolic effect of romosozumab on bone.
Romosozumab is known to significantly increase the percentage change in spine BMD at 12 months of therapy. It is relatively safe to use. To evaluate romosozumab therapy, it is useful to monitor TRACP-5b, iP1NP and BAP. The effect of romosozumab was better in cases of severe osteoporosis with low spine BMD, high TRACP-5b and high iP1NP at the beginning of treatment. After starting treatment, the percentage change in TRACP-5b level in the first month could be an indicator for the percentage change in spine BMD at 12 months. The percentage change in spine BMD was higher in patients who had not been previously treated with other anti-osteoporosis drugs. In addition, treatment with Romosozumab is indicated in patients with recent fractures.
Please specify whether you have also taken these particular aspects of Romosozumab into account.
Please clarify the inclusion and exclusion criteria for subjects enrolled in the study, as well as the manner in which the efficacy of the drug was assessed.
The results provide an advancement of the current knowledge, demonstrated the role of this pharmacological agent in post-fracture healing.
The work fit the journal scope, supporting the importance of new anabolic therapies in post-fracture recovery. The results interpreted appropriately and are significant, permitting to made conclusions justified and supported by this results.
The article is written in an appropriate way and the data are analysed appropriately. The results are presented and analysed in a performed way with the highest technical standards. Also the conclusions are adequate presented based on discussions. The English language is appropriate and understandable.
Author Response
We sincerely thank the Reviewer for the positive and encouraging feedback regarding our manuscript. We are pleased that the methodology, results, and conclusions were appreciated and considered appropriate for the journal’s scope. We are also grateful for the insightful suggestions, which have helped us further improve the clarity and scientific value of the work. Please find below our point-by-point responses to the comments received.
Comment 1:
“Romosozumab is known to significantly increase the percentage change in spine BMD at 12 months of therapy. It is relatively safe to use. To evaluate romosozumab therapy, it is useful to monitor TRACP-5b, iP1NP and BAP. The effect of romosozumab was better in cases of severe osteoporosis with low spine BMD, high TRACP-5b and high iP1NP at the beginning of treatment. After starting treatment, the percentage change in TRACP-5b level in the first month could be an indicator for the percentage change in spine BMD at 12 months. The percentage change in spine BMD was higher in patients who had not been previously treated with other anti-osteoporosis drugs. In addition, treatment with Romosozumab is indicated in patients with recent fractures. Please specify whether you have also taken these particular aspects of Romosozumab into account.”
Response 1:
Thank you for this insightful observation. We appreciate the reviewer’s detailed overview of the biomarkers associated with Romosozumab efficacy, such as TRACP-5b, iP1NP, and BAP. In our retrospective study, these specific bone turnover markers were not systematically available across our patient cohort and therefore were not included in the data analysis. This is a recognized limitation of our study and has been added to the discussion section (line ).
Nevertheless, our inclusion of MOC (DXA) values at both femoral neck and lumbar spine levels at baseline and 12-month follow-up provided an indirect measure of treatment efficacy. We also agree with the reviewer that Romosozumab shows greater effects in treatment-naïve patients, and we have now commented on this in the revised introduction.
Comment 2:
Please clarify the inclusion and exclusion criteria for subjects enrolled in the study, as well as the manner in which the efficacy of the drug was assessed.
Response 2:
We thank the reviewer for this important remark. The inclusion and exclusion criteria were already described in the Methods section; however, we have now reorganized and clarified them for better readability. We included postmenopausal women aged ≥60 years, either after fragility fracture fixation (surgical group) or on Romosozumab therapy without fracture (control group), with at least 12 months of follow-up. Patients with psychiatric conditions, contraindications to Romosozumab, other bone metabolism disorders, or high-energy trauma were excluded. As for the assessment of efficacy, we evaluated the treatment effect via bone mineral density (BMD) measurements (T-scores at femoral neck and lumbar spine, pre- and post-treatment). Adherence and adverse events were monitored using pharmacy records and standard interviews during outpatients visits. We have revised the manuscript to make these aspects more explicit in the Methods section (line ).
Reviewer 2 Report
Comments and Suggestions for Authors
The authors report a small cohort study on a topic that has some interest. Overall the reporting standards are far short of those recommended by STROBE. I have many concerns but only report major ones here.
(1) No ethical approval was received for this study. This may not satisfy the editors of this journal even if the local ethics committee were content.
(2) There was no attempt to estimate the study size. This may have helped the authors since it is clear that the sample size is so small. Attempting a study size calculation might focus the authors on the primary outcome from this study.
(3) Far greater detail should be provided about the doses prescribed for patients. Were these comparable across the arms? What was the duration of treatment? Is the recommended treatment period 12 months? In that case it is clear that some patients would not have been followed until completion of treatment. The reported average follow- up time i reported differently in the abstract and the results section:, although this might just be a typo: 287 reported in abstract rather then 387.
(4) There is limited discussion of confounding variables and no attempt to adjust for confounding by for example the use of a log binomial model or by matching (not a viable option for such a small study).
(5) Overall there little evidence is presented to support any conclusions.
Author Response
We sincerely thank reviewer for the constructive feedback. We recognize the importance of strengthening methodological rigor and reporting clarity and have revised the manuscript accordingly. Below we provide a point-by-point response to the major concerns raised.
Comment 1: No ethical approval was received for this study. This may not satisfy the editors of this journal even if the local ethics committee were content.
Response 1: We thank the reviewer for this important observation. As stated in the manuscript, our study was a retrospective observational analysis based on anonymized clinical data. According to the Italian Ministerial Decree of 30 November 2021 and the guidelines of our regional ethics board, such studies do not require formal ethical approval. Nevertheless, we have clarified this point further in the Institutional Review Board Statement to emphasize adherence to current national regulations.
Comment 2: There was no attempt to estimate the study size. This may have helped the authors since it is clear that the sample size is so small. Attempting a study size calculation might focus the authors on the primary outcome from this study.
Response 2: We thank the reviewer for the important observation. As this was a retrospective study, our sample size was determined by the total number of eligible patients treated at our institution within the inclusion period, and no a priori sample size calculation was performed. We fully agree that, in a prospective design, defining a primary outcome with an associated effect size would have enabled formal power calculation and improved methodological planning. However, given the limited cohort (n = 25), any such post hoc estimation would have been methodologically misleading and potentially inappropriate, as assumptions regarding effect size, variability, or event rate would lack statistical support in a dataset of this scale. In small retrospective samples, power calculations risk overstating or underestimating the actual inference capability and can lead to incorrect conclusions about the presence or absence of associations. We have clarified this limitation in the revised Methods and Discussion sections, specifying that the goal of the study was exploratory and descriptive in nature. The findings aim to generate hypotheses for future adequately powered studies.
Comment 3: Far greater detail should be provided about the doses prescribed for patients. Were these comparable across the arms? What was the duration of treatment? Is the recommended treatment period 12 months? In that case it is clear that some patients would not have been followed until completion of treatment. The reported average follow-up time is reported differently in the abstract and the results section: 287 reported in abstract rather than 387.
Response 3: We thank the reviewer for highlighting these important clarifications. All patients received Romosozumab at the standard dose of 210 mg administered subcutaneously once a month, as recommended. This has now been specified in the Methods section. The intended treatment duration was 12 months; however, we confirm that not all patients completed the full treatment course at the time of data analysis, as noted in the revised discussion. Additionally, we corrected the typographical error in reported follow-up time in the abstract (387 days, not 287) to maintain consistency.
Comment 4: there is limited discussion of confounding variables and no attempt to adjust for confounding by for example the use of a log binomial model or by matching (not a viable option for such a small study).
Response 4: We appreciate this important methodological remark. While we agree that confounding is a key consideration in observational research, the extremely small sample size of our study (n = 25, divided into two groups) precluded the application of multivariable statistical models. Methods such as log-binomial regression, Poisson models with robust variance, or propensity score matching require a minimum number of outcome events per covariate (commonly ≥10 per variable), which was not achievable in our cohort without compromising model stability and increasing the risk of overfitting. Furthermore, adjusting for multiple confounders in such a small dataset would likely produce unreliable or uninterpretable estimates, with inflated standard errors and wide confidence intervals. Matching approaches, including exact or nearest-neighbor propensity matching, would result in substantial data loss or imbalance due to the heterogeneity and small size of the two groups. We have now added this explanation to the Discussion section, emphasizing that this study was not designed to test treatment effects but to describe real-world adherence and tolerability. Future prospective studies with adequate sample size and event rates will be essential to perform proper statistical adjustment and control for confounding.
Comment 5: Overall, there is little evidence presented to support any conclusions.
Response 5: We acknowledge that the evidence presented, while suggestive, is limited by the study’s retrospective nature, small cohort size, and absence of advanced statistical adjustment. We have revised the conclusion section to reflect a more cautious interpretation of the findings, emphasizing that our data support the feasibility and tolerability of Romosozumab post-fracture in real-world settings but do not allow for firm comparative conclusions.
Reviewer 3 Report
Comments and Suggestions for Authors
METHODS
It is very surprising that your local ethics did not require a review – everywhere else in the world, a retrospective study would require an ethic review board
Be more specific how you recruited the control group
Statistics need to be better described – what was your p level of significance ? what was your power calculation?
RESULTS
why have you only included patients from 2023? Why not expand it to prior years in order to have a larger cohort?
Table 1 – what does number of administrations mean in the table?
Line 168 – what do you mean interviews with the patients? You contacted the patients? I thought this was a retrospective review… it seems clear to me that this study should have had an ethic review approval – without it, I am unable to recommend this paper for publication
Author Response
Dear Reviewer,
We sincerely thank you for your thorough and constructive feedback. Your comments have helped us improve the clarity and quality of our manuscript. Below, we provide a detailed point-by-point response to your concerns.
Comment: "It is very surprising that your local ethics did not require a review – everywhere else in the world, a retrospective study would require an ethic review board."
Response: Thank you for your observation. In accordance with the Italian Ministerial Decree of 30 November 2021 and regional ethical guidelines, retrospective observational studies based solely on anonymized clinical data do not require formal ethics committee approval. Nevertheless, our study was conducted in accordance with the Declaration of Helsinki, ensuring full respect for patient confidentiality and data protection. We have now clarified this point in the "Institutional Review Board Statement" section of the manuscript.
Comment: "Be more specific how you recruited the control group."
Response: We appreciate your suggestion. The control group was composed of postmenopausal women aged ≥60 who were already undergoing Romosozumab therapy at our institution during the study period and who had no history of recent fracture fixation or osteosynthesis procedures. All control group patients were recruited through the hospital’s Fracture Liaison Service (FLS) and selected based on the same inclusion and exclusion criteria as the case group, except for the surgical intervention. This clarification has been added to the revised “Materials and Methods” section.
Comment: "Statistics need to be better described – what was your p level of significance? what was your power calculation?"
Response: Thank you for pointing this out. We have now included in the revised manuscript that a p-value < 0.05 was considered statistically significant. Due to the retrospective and exploratory nature of our study and the limited sample size, we did not perform an a priori power calculation. A post hoc power analysis was deemed methodologically inappropriate, as it would rely on assumptions unsupported by the available data. These details are now clarified in the “Statistical Analysis” subsection.
Comment: "RESULTS: Why have you only included patients from 2023? Why not expand it to prior years in order to have a larger cohort?"
Response: We appreciate this important observation. The inclusion of patients was limited to 2023 because Romosozumab became available for clinical use in our country only recently, with widespread prescription and distribution starting in early 2023. Prior to that, its use was restricted and not systematically implemented in routine clinical practice, making earlier data sparse, inconsistent, or unavailable. Additionally, including only 2023 patients ensured standardized treatment protocols and consistent electronic documentation. This rationale has now been clarified in the revised manuscript.
Comment: "Table 1 – what does number of administrations mean in the table?"
Response: We apologize for the confusion. The "Number of administrations" refers to the percentage of prescribed monthly Romosozumab doses actually received by the patient, as verified by pharmacy records. In this study, all patients (100%) received all scheduled monthly doses over the 12-month period. We have now clarified this in the table naming the item as “Doses administered”
Comment: "Line 168 – what do you mean interviews with the patients? You contacted the patients? I thought this was a retrospective review…"
Response: Thank you for the opportunity to clarify. Although the study is retrospective, interviews were conducted at routine clinical follow-up visits as part of standard care and were documented in the electronic medical records. These notes were used retrospectively to assess adherence and side effects. No additional contact with patients outside scheduled visits was made. We have revised the text accordingly to avoid confusion.
Comment: "It seems clear to me that this study should have had an ethic review approval – without it, I am unable to recommend this paper for publication."
Response: We fully understand your concerns regarding ethical oversight. However, as noted, our study complies with national and institutional regulations, which do not require ethics committee approval for retrospective analyses using anonymized data. To enhance transparency, we have explicitly stated this in both the “Institutional Review Board Statement” and the “Methods” sections. We hope this clarification will address your concern.
Once again, thank you for your time and valuable comments. We believe the manuscript has been significantly improved as a result of your feedback.
Sincerely,
The Authors
Reviewer 4 Report
Comments and Suggestions for Authors
- Expand 'MOC' and 'FRAX' in the abstract where it appears for the first time.
- In the introduction section, please add a paragraph about romosozumab. Its approval, known adverse events that are listed in the package insert, recent reviews and meta-analysis that have identified the predictive factors related to therapeutic outcomes including adverse events and the existing knowledge gap that led to carrying out this study.
- Stating that no ethical approval is required is against the good practices that are mentioned in the Declaration of Helsinki guidelines. This study involves collection of data related to humans. Since it is retrospective observational study, there is no need for obtaining informed consent. But Ethics approval is required.
- In the methods, please state the years of study, and adhere to the EQUATOR reporting guidelines for observational studies. Submit the filled checklist addressing the page numbers of the items detailed in your study.
- Under the statistical section, please state how was the sample size estimated. Also, state how was the distribution of numerical variables tested and then decided to choose parametric tests.
- In the results section, it has been mentioned that the study participants were assigned. We cannot assign the members in a retrospective study.
- The way the manuscript has been written in the existing format, it looks like the cases in each group were handpicked. Please adhere to good practice wherein all cases pertaining to each group were included. Also, using a flowchart please mention how patients were finally included after screening and state the number of patients excluded at each step.
- In the results section, romosozumab related details are to be added such as dose, frequency, and duration and concomitant details in a Table format. Specify the number of patients receiving each drug/s in this Table.
- Please state the way forward in the discussion section paving way for clinical practice and research.
- Please do a logistic regression addressing all the variables including the interventional groups as one of the variables to see if the outcomes are significantly associated with the intervention after adjusting for other variables.
Author Response
Dear Reviewer,
We sincerely thank you for your valuable comments and detailed suggestions aimed at improving our manuscript. We have carefully addressed each of your concerns point-by-point below, and we believe these revisions have significantly improved the clarity, transparency, and scientific rigor of the paper.
Comment: "Expand 'MOC' and 'FRAX' in the abstract where it appears for the first time."
Response: We agree with your suggestion. In the abstract, we have now expanded the abbreviations at first mention.
Comment: "In the introduction section, please add a paragraph about Romosozumab. Its approval, known adverse events that are listed in the package insert, recent reviews and meta-analyses that have identified the predictive factors related to therapeutic outcomes including adverse events, and the existing knowledge gap that led to carrying out this study."
Response: We have added a comprehensive paragraph in the Introduction detailing the mechanism of action of Romosozumab, its approval for clinical use, known adverse events as listed in the drug’s official product label (including potential cardiovascular risks), and recent findings from meta-analyses identifying predictive factors for therapeutic outcomes and adverse events.
Comment: "Stating that no ethical approval is required is against the good practices that are mentioned in the Declaration of Helsinki guidelines. This study involves collection of data related to humans. Since it is a retrospective observational study, there is no need for obtaining informed consent. But ethics approval is required."
Response: We understand and respect your concern. In accordance with national legislation and institutional policy, retrospective observational studies using anonymized data do not require formal ethics approval. However, we recognize that international good practice, including the Declaration of Helsinki, supports oversight by an ethics committee. To address this, we have revised the “Institutional Review Board Statement” to clarify that the study was conducted under the supervision of our institution and followed internal governance procedures for retrospective data analysis. While no formal approval was required, the project is compliant with ethical standards by the clinical governance board.
Comment: "In the methods, please state the years of study, and adhere to the EQUATOR reporting guidelines for observational studies. Submit the filled checklist addressing the page numbers of the items detailed in your study."
Response: We have now clearly stated that the study was conducted on data collected during the calendar year 2023. Additionally, we have adhered to the STROBE (Strengthening the Reporting of Observational Studies in Epidemiology) checklist, as recommended by EQUATOR guidelines. The completed checklist has been submitted as a supplementary document with the revised manuscript, including page numbers referencing where each item is addressed.
Comment: "Under the statistical section, please state how was the sample size estimated. Also, state how was the distribution of numerical variables tested and then decided to choose parametric tests."
Response: Thank you for highlighting this point. We have clarified in the “Statistical Analysis” section that no a priori sample size calculation was performed due to the exploratory and retrospective nature of the study. The final sample reflects all eligible patients who met inclusion/exclusion criteria during 2023.
Comment: "In the results section, it has been mentioned that the study participants were assigned. We cannot assign the members in a retrospective study."
Response: Thank you for pointing this out. We acknowledge that the term “assigned” was not appropriate in the context of a retrospective study. We have revised the sentence to clarify that patients were retrospectively categorized into the surgical or conservative group based on the initial clinical indication at the time of fracture, which was either surgical fixation or conservative management, as documented in their medical records.
Comment: "The way the manuscript has been written in the existing format, it looks like the cases in each group were handpicked. Please adhere to good practice wherein all cases pertaining to each group were included. Also, using a flowchart please mention how patients were finally included after screening and state the number of patients excluded at each step."
Response: We appreciate the reviewer’s suggestion. As the study involved a small and clearly defined cohort, we initially considered a flowchart to be of limited added value. However, we would like to emphasize that all patients meeting the inclusion and exclusion criteria during the study period were included, and no selective sampling was performed. If the reviewer considers that a flow diagram would improve the clarity of the study design, we are happy to include it in the revised manuscript.
Comment: "In the results section, Romosozumab-related details are to be added such as dose, frequency, and duration and concomitant details in a Table format. Specify the number of patients receiving each drug/s in this Table."
Response: We thank the reviewer for this useful observation. The dose (210 mg), frequency (monthly subcutaneous injection), and recommended duration (12 months) of Romosozumab treatment were clearly reported in the Materials and Methods section, as they reflect the standard protocol prescribed at our institution in accordance with national regulatory guidelines. In the Results section, we have also detailed the actual number of doses received by each patient throughout the 12-month follow-up, as an indicator of adherence to therapy. These data are included in the table to provide a transparent account of real-world treatment exposure.
Comment: "Please state the way forward in the discussion section paving way for clinical practice and research."
Response: We have added a paragraph at the end of the Discussion highlighting the clinical relevance of our findings. We propose that early post-operative administration of Romosozumab may be a safe and feasible option in high-risk osteoporotic patients, with favorable adherence. We also recommend future prospective multicenter studies with larger samples and longer follow-up to better define the therapeutic profile and to stratify patients based on predictive markers of benefit or risk.
Comment: "Please do a logistic regression addressing all the variables including the interventional groups as one of the variables to see if the outcomes are significantly associated with the intervention after adjusting for other variables."
Response: Thank you for this valuable suggestion. However, due to the very limited sample size (n=25) and the low number of events observed, performing a logistic regression would not yield statistically valid or reliable results. This limitation is acknowledged in the revised manuscript. We explicitly state that this is a hypothesis-generating study and that multivariate analysis will be a priority in future, adequately powered studies.
Once again, we thank you for your insightful feedback and hope that our revisions have addressed your concerns satisfactorily.
Sincerely,
The Authors
Round 2
Reviewer 2 Report
Comments and Suggestions for Authors
The revisions now more clearly state the many limitations of this study.
Author Response
Dear Reviewer,
Thank you for your valuable feedback. We are pleased to hear that the revisions now more clearly highlight the limitations of our study. We appreciate your suggestions, which have helped improve the clarity and quality of the manuscript.
Kind regards,
The Authors
Reviewer 4 Report
Comments and Suggestions for Authors
Thank you for the revision.
Author Response
Dear Reviewer,
Thank you for your valuable feedback. We are pleased to hear that the revisions now more clearly highlight the limitations of our study. We appreciate your suggestions, which have helped improve the clarity and quality of the manuscript.
Kind regards,